# PariedContrast: A Multimodal Benchmark for Contrast Image Translation

## Abstract

Contrast medium play a pivotal role in radiological imaging, as it amplifies lesion conspicuity and improves detection in the diagnosis of tumor-related diseases. However, depending on the patient's health condition or the medical resources available, the use of contrast medium is not always feasible. Recent work has therefore explored AI-based image translation to synthesize contrast-enhanced images directly from non-contrast scans, aiming to reduce side effects and streamline clinical workflows. Progress in this direction has been constrained by data limitations: (1) existing public datasets focus almost exclusively on brain-only paired MR modalities; (2) other collections include partially paired data but suffer from missing modalities/timestamps and imperfect spatial alignment; (3) explicit labeling of CT vs. CTC or DCE phases is often absent; (4) substantial resources remain private. To bridge this gap, we introduce the first public, fully paired, pan-cancer medical imaging dataset spanning 11 human organs. The MR data include complete dynamic contrast-enhanced (DCE) sequences covering all three phases (DCE1–DCE3), while the CT data provide paired non-contrast and contrast-enhanced acquisitions (CTC). The dataset is curated for anatomical correspondence, enabling rigorous evaluation of $1 \rightarrow 1$, $N \rightarrow 1$, and $N \rightarrow N$ translation settings (e.g., predicting DCE phases from non-contrast inputs). Built upon this resource, we establish a comprehensive benchmark. We report results from representative baselines of contemporary image-to-image translation. We release the dataset and benchmark to catalyze research on safe, effective contrast synthesis, with direct relevance to multi-organ oncology imaging workflows.

## 1 Introduction

Accurate diagnosis from medical images often depends on how clearly subtle tissue differences can be visualized. Contrast medium amplifies these differences by highlighting lesions and vascular structures, thereby improving the sensitivity and reliability of tumor detection. Yet their use is not always feasible: certain patients face health risks from contrast administration, and resource limitations can further restrict availability. Consequently, many scans are acquired without contrast enhancement, leaving clinicians with incomplete diagnostic information.

Recent advances in generative AI (Heusel et al., 2017; Rombach et al., 2022; Peebles & Xie, 2023) offer promising solutions by synthesizing contrast-enhanced images directly from non-contrast scans (Atli et al., 2024; Chartsias et al., 2018). Such approaches open the door to safer imaging protocols and streamlined clinical workflows. However, their development critically depends on access to large-scale, well-curated paired datasets spanning diverse organs and cancer types.

Existing resources remain inadequate. (1) Publicly available paired MRI datasets, such as BraTS (de Verdier et al., 2024), are almost exclusively limited to brain imaging. (2) Other collections, including AMOS (Ji et al., 2022) or datasets from TCIA (Clark et al., 2013), provide partially paired CT and MRI data but suffer from missing modalities, timestamps, or imperfect spatial alignment. (3) Explicit annotations distinguishing non-contrast CT from contrast-enhanced CT (CTC) or delineating dynamic contrast-enhanced (DCE) MRI phases are often absent, as in CT-ORG (Rister et al., 2020). (4) Finally, substantial multi-organ resources cohorts remain private, limiting community-wide benchmarking.

Figure 1: Representative task settings with examples. (a) CT → CTC (1→1), (b) DCE$_1$ → DCE2 (1→1), (c) DCE1, 3 → DCE$_2$ (N→1), (d) DCE1 → DCE2, 3 (1→N).

To bridge this gap, we introduce PariedContrast, the first *public, fully paired, pan-cancer* dataset covering 11 human organs. It provides complete dynamic contrast-enhanced MRI sequences (DCE1–DCE3), alongside paired non-contrast and contrast-enhanced CT (CTC). All data are carefully curated for anatomical correspondence, enabling systematic evaluation of image translation tasks under $1 \to 1$, $N \to 1$, and $N \to N$ settings.

Building upon this resource, we establish a comprehensive benchmark by evaluating representative baseline methods in image-to-image translation, including GAN-based and diffusion-based models. Beyond these baselines, we introduce FlowMI, a flow-based missing modality imputation model inspired by recent advances in latent flow matching. Instead of substituting missing inputs with zeros or noise, FlowMI projects both complete and incomplete modalities into a shared latent space, via modality-specific encoders combined through a product-of-experts aggregation. It then learns a continuous flow that transforms the resulting "broken" latent codes to their fully observed counterparts. This design allows robust reconstruction under arbitrary missing patterns and achieves superior performance in recovering details. Notably, the ability to recover fine-grained details is critical for downstream clinical tasks such as tumor detection. Together, the dataset, benchmark, and FlowMI establish a strong foundation for advancing safe and effective contrast synthesis. Our key contributions are summarized as follows:

1. We present the first *public*, *fully paired* contrast-enhanced and non-contrast, *pan-cancer* dataset, providing a large-scale, high-quality resource to facilitate medical imaging research.
2. We propose FlowMI, a flow-matching model that captures complex cross-modality relationships and leverages an uncertainty mitigation strategy, leading to more reliable multimodal image synthesis with better potential for downstream clinical applications.
3. We conduct a comprehensive benchmark across diverse organs, modalities, and translation tasks, establishing strong reference results for future research. Our proposed FlowMI achieves consistently superior performance across settings.

## 2 THE BENCHMARK

In this section, we first define the task enabled by PariedContrast (section 2.1). We then describe the data curation and preparation process (section 2.2), followed by a quantitative analysis of the quality and diversity of PariedContrast (section 2.3). Finally, we compare our dataset against existing benchmarks (section 2.4) to highlight its unique advantages.

### 2.1 TASK DEFINATION

Generative models for medical image translation aim to model complex anatomical structures, capture modality-specific features, and learn accurate cross-modality mappings. Formally, given a set of input modalities $X = \{x_1, x_2, \ldots, x_n\}$ and one or more target modalities $Y = \{y_1, y_2, \ldots, y_m\}$, a generative model $f$ produces synthesized images $\hat{Y}$ that approximate the ground-truth targets $Y$:

$$\hat{Y} = f(X). \tag{1}$$

In this study, we consider several widely used imaging modalities: Computed Tomography (CT), contrast-enhanced CT (CTC), and multiple Magnetic Resonance Imaging (MRI) sequences, including Dynamic Contrast-Enhanced MRI (DCE).

To comprehensively evaluate different generative models, we design benchmark tasks that reflect both increasing levels of difficulty and common clinical scenarios of missing modalities. As illustrated in Fig. 1, we consider three representative settings:

Table 1: Detailed statistics of the PairedContrast dataset.

| Modality | Overall of Modality | Type of Modality | Dataset | System | Overall of Dataset | Organ | Overall of Organ | Source Dataset | Overall of Dataset | Type of Cancer |
|---|---|---|---|---|---|---|---|---|---|---|
| MR | 1116 | dce1, dce2, dce3 | Breast_MR_train_val_test | Female Reproductive | 1116 | Breast | 1116 | TCGA-BRCA / UCSF / I-SPY 1 | 378 / 180 / 558 | Breast Cancer |
| CT | 1526 | Non-contrast CT, Contrast-enhanced CT | Adrenal_CT_train_val_test | Endocrine | 82 | Adrenal | 82 | Adrenal-ACC-Ki67-Seg | 82 | Adrenocortical carcinoma |
| | | | Uterus_Ovary_CT_train_val_test | Female Reproductive | 66 | Ovary | 12 | TCGA-OV | 12 | Ovarian Serous Cystadenocarcinoma |
| | | | | | | Uterus | 54 | TCGA-UCEC | 14 | Uterine Corpus Endometrial Carcinoma |
| | | | | | | | | CPTAC-UCEC | 40 | |
| | | | Stomach_Colon_Liver_Pancreas_CT_train_val_test | Digestive | 614 | Stomach | 86 | TCGA-STAD | 86 | Stomach Adenocarcinoma |
| | | | | | | Pancreas | 54 | CPTAC-PDA | 54 | Ductal Adenocarcinoma |
| | | | | | | Liver | 432 | HCC-TACE-Seg | 360 | Hepatocellular carcinoma |
| | | | | | | | | TCGA-LIHC | 72 | |
| | | | | | | Colon | 22 | CMB-CRC | 18 | Colorectal Cancer |
| | | | | | | | | TCGA-COAD | 4 | Colon adenocarcinoma |
| | | | Bladder_Kidney_CT_train_val_test | Urinary | 684 | Bladder | 86 | TCGA-BLCA | 86 | Bladder Endothelial Carcinoma |
| | | | | | | Kidney | 598 | TCGA-KIRC | 278 | Kidney Renal Clear Cell Carcinoma |
| | | | | | | | | C4KC-KiTS | 216 | Kidney Cancer |
| | | | | | | | | CPTAC-CCRCC | 66 | Clear Cell Carcinoma |
| | | | | | | | | TCGA-KIRP | 26 | Kidney Renal Papillary Cell Carcinoma |
| | | | | | | | | TCGA-KICH | 12 | Kidney Chromophobe |
| | | | | | | | | CMB-LCA | 28 | Lung Cancer |
| | | | Lung_CT_train_val_test | Respiratory | 80 | Lung | 80 | Lung-PET-CT-Dx | 6 | |
| | | | | | | | | Anti-PD-1_Lung | 6 | |
| | | | | | | | | TCGA-LUSC | 2 | Lung Squamous Cell Carcinoma |
| | | | | | | | | CPTAC-LSCC | 28 | Squamous Cell Carcinoma |
| | | | | | | | | CPTAC-LUAD | 10 | Lung Adenocarcinoma |

**1-to-1 Translation:** Single input modality to a single target (*e.g.*, CT → CTC or DCE$_1$ → DCE$_2$). This setting tests a model's ability to capture modality-specific features and preserve one-to-one anatomical correspondences.

**N-to-1 Translation:** Multiple input modalities to a single target (*e.g.*, DCE$_1$, DCE$_3$ → DCE$_2$). This evaluates how well models integrate complementary anatomical information across sequences while maintaining structural fidelity and modality consistency. In DCE imaging, this setting further corresponds to reconstructing an intermediate phase from its neighbors, thereby probing whether models can capture temporal dynamics of contrast uptake.

**1-to-N Translation:** A single input modality to multiple targets simultaneously (*e.g.*, DCE$_1$ → DCE$_2$, DCE$_3$). This setting assesses whether models can jointly capture inter-modal dependencies and generate anatomically consistent outputs across domains. Clinically, it is relevant for scenarios where only an early phase or a non-contrast scan is acquired, and later phases must be synthesized to approximate the full dynamic sequence.

Together, these tasks can be unified under the general **N-to-N translation** formulation, where both input and output may consist of multiple modalities. They span a spectrum from fundamental to highly challenging scenarios, ensuring that our benchmark is representative of real-world clinical requirements for missing modality synthesis.

## 2.2 DATASET CURATION AND PREPARATION

A major obstacle for contrast synthesis research is the lack of large-scale, paired datasets spanning multiple organs and imaging modalities. To address this, we curated PariedContrast, a multi-organ, pan-cancer resource constructed entirely from publicly available sources.

PariedContrast distinguishes itself by providing: (i) explicit pairing of contrast-enhanced (CE) and non-contrast-enhanced (NCE) scans across both CT and MRI, and (ii) broad coverage of clinically relevant organs frequently encountered in oncology. As illustrated in Fig. 2, the curation process consists of dataset selection, quality filtering, and standardized preprocessing to ensure anatomical correspondence and clinical validity.

**Targeted organs and imaging modalities.** Based on three criteria—(i) availability of paired CE/NCE scans, (ii) prevalence in oncology imaging studies, and (iii) the clinical importance of con-

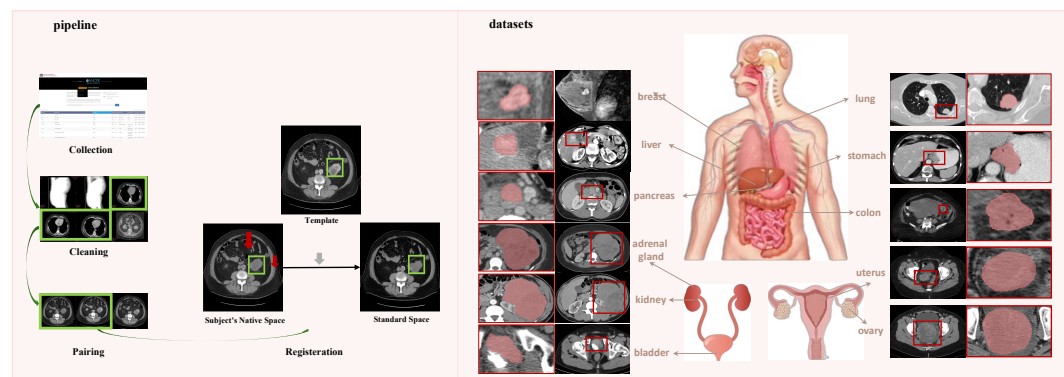

Figure 2: *Left:* dataset curation pipeline. *Right:* representative examples from PariedContrast.

trast enhancement for lesion delineation—we selected 11 organs spanning both CT(adrenal gland, ovary, uterus, stomach, pancreas, liver, colon, bladder, kidney, lung) and MRI(breast).

**Data selection and quality filtering.** All scans were sourced from publicly available repositories such as The Cancer Imaging Archive (TCIA) and related datasets (*e.g.*, TCGA, CPTAC, CMB, UCSF, I-SPY). We identified collections with paired CE/NCE scans or reliable metadata indicating contrast phase, retrieved the raw DICOM/NIfTI files, and excluded studies with severe artifacts, incomplete coverage, or corrupted files. This multi-stage filtering ensured that only scans with reliable contrast information and sufficient anatomical coverage were retained for preprocessing.

**Preprocessing with clinical validation.** To harmonize diverse sources into a unified benchmark, we applied a standardized pipeline:

*Contrast-pair identification:* automated metadata parsing (SeriesDescription, ContrastBolusAgent, AcquisitionTime) was used to distinguish CE/NCE scans.
*Validation and quality control:* trained annotators verified preliminary CE/NCE modalities and overall image quality, discarding ambiguous or low-quality cases. Final confirmation was provided by experts, who explicitly labeled CT *vs.* CTC and identified DCE time steps.
*Registration:* rigid and affine alignment of CE and NCE scans, with deformable registration for motion-prone organs (e.g., liver, lung).
*Cropping and resampling:* organ-level bounding box cropping and resampling to isotropic spacing (1×1×1 mm).
*Normalization:* For CT, hounsfield Unit (HU) windowing per organ (*e.g.*, soft tissue: $[-200, 300]$ HU), then min–max normalization. For MRI, z-score normalization per scan to mitigate inter-patient intensity variation.
*Pairing verification:* final visual inspection to ensure anatomical correspondence between CE and NCE scans.

## 2.3 DATA STATISTICS AND DIVISION

The PariedContrast collection is organized hierarchically: first by imaging modality (Magnetic Resonance, MR; Computed Tomography, CT), and then by organ-specific groups. In total, the dataset spans 11 organs grouped into 6 subsets representing 5 anatomical systems: Endocrine (adrenal), Digestive (stomach, pancreas, liver, colon), Urinary (bladder, kidney), Respiratory (lung), and Female Reproductive (uterus, ovary, and breast).

Across these organs, the dataset covers 19 cancer types, including: adrenocortical carcinoma (adrenal); ovarian serous cystadenocarcinoma (ovary); uterine corpus endometrial carcinoma (uterus); stomach adenocarcinoma (stomach); ductal adenocarcinoma and hepatocellular carcinoma (liver); colon adenocarcinoma and colorectal cancer (colon); bladder urothelial carcinoma (bladder); kidney renal clear cell, papillary cell, and chromophobe carcinomas (kidney); lung adenocarcinoma and squamous cell carcinoma (lung); and breast carcinoma (breast). A detailed breakdown of case numbers per organ, modality, and cancer type is provided in Appendix.

For benchmarking, the dataset is split into training (70%), validation (10%), and test (20%) sets. Stratified sampling ensures proportional representation across organ systems and cancer types. The

Table 2: A comparison of our proposed PairedContrast to other benchmarks.

| Benchmarks | Organ | MR | CT | Paired | Size | Application |
|---|---|---|---|---|---|---|
| CHAOS(Valindria et al., 2018) | Abdomen | ✓ | ✓ | | 120 | Healthy abdominal organ research |
| BraTS 2025 (Maleki et al., 2025) | Brain | ✓ | | ✓ | 4425 | Brain tumor research |
| IXI (IXI) | Brain | ✓ | | | 600 | Healthy brain research |
| crossMoDA (Dorent et al., 2023) | Ear | ✓ | | | 379 | Cochlear implant research |
| ACDC (Bernard et al., 2018) | Cardiac | ✓ | | | 150 | Cardiac diagnosis |
| MMWHS(Zhuang, 2018) | Cardiac | ✓ | ✓ | | 120 | Cardiac research |
| FDG-PET/CT (Gatidis et al., 2022) | Whole body | ✓ | ✓ | | 1014 | Tumor, lung cancer research |
| **Ours** | Adrenal, Breast, Bladder, Colon, Kidney, Liver, Lung, Ovary, Pancreas, Stomach, Uterus | ✓ | ✓ | ✓ | 2642 | Pan-cancer research |

test set is further subdivided into a *test-mini* split (5% of the full dataset), designed for rapid validation while preserving the distribution of the complete test set. Unless otherwise specified, all reported results are based on the *test-mini* split.

## 2.4 COMPARISONS WITH EXISTING BENCHMARKS

We position PariedContrast against representative public benchmarks along five axes relevant for multimodal image translation and missing-modality synthesis: (i) organ/system coverage, (ii) imaging modalities (MR/CT and contrast availability), (iii) explicit CE–NCE pairing at the *per-patient* level, (iv) dataset scale and balance, and (v) primary application focus. Tab. 2 summarizes the comparison.

**Brain-centric MR benchmarks.** Resources such as BraTS (de Verdier et al., 2024), BraSyn (Li et al., 2024), IXI (IXI), OASIS-3 (LaMontagne et al., 2019), and ADNI (Rivera Mindt et al., 2024) provide large-scale brain MRI data, supporting tumor segmentation and neurodegeneration studies. However, they are *single-organ* and typically lack explicit CE–NCE pairing, limiting their applicability to pan-organ contrast translation.

**Cardiac and region-specific benchmarks.** Datasets such as ACDC (Bernard et al., 2018) and MMWHS (Zhuang, 2018) offer high-quality cardiac MR/CT for segmentation and multi-modality analysis. Similarly, crossMoDA (Dorent et al., 2023) focuses on inner-ear/temporal-bone MRI for domain adaptation. These are *task-focused and organ-specific* resources without systematic CE–NCE pairing, hence not tailored for generalizable contrast synthesis across diverse organs.

**PET/CT and dose-constrained benchmarks.** Whole-body PET/CT sets (e.g., FDG-PET/CT (Gatidis et al., 2022)) and ultra-low-dose PET studies (UDP, 2024) target cross-modality fusion or dose reduction and often assess PET synthesis from limited counts. While multi-modality is present, CE–NCE pairing for CT/MR contrast translation is typically outside their scope.

**Positioning of PariedContrast.** In contrast, PariedContrast (*PairedContrast*) provides explicit per-patient CE–NCE pairing across 11 organs spanning both CT and MR. It is designed for multimodal translation and missing-modality synthesis, with harmonized preprocessing and radiologist-in-the-loop validation. As shown in Tab. 2, most existing benchmarks emphasize either a single organ (*e.g.*, brain, cardiac) or tasks orthogonal to contrast translation (*e.g.*, PET dose reduction). PariedContrast fills this gap by offering multi-organ breadth, dual-modality coverage, and rigorously verified CE–NCE pairs—enabling clinically meaningful benchmarking of contrast synthesis methods.

## 3 METHODS

### 3.1 PRELIMINARY: LATENT FLOW MATCHING

Flow Matching (FM) (Lipman et al., 2023; Liu et al., 2022) learns a continuous transport map between two distributions over a data space $\mathcal{X} \subseteq \mathbb{R}^d$, using only samples and without access to

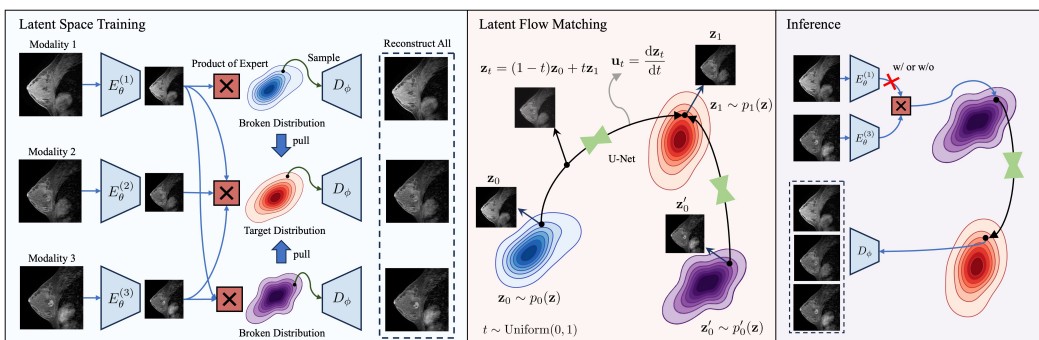

Figure 3: Overview of the proposed FlowMI framework. **Left:** Modality-specific encoders $E_\theta^{(i)}$ map inputs into a latent space, which are fused via a product-of-experts. The distribution with all modalities defines the *target*, while cases with missing modalities define the *broken* distribution. **Middle:** Latent flow matching learns a smooth mapping from $p_0(z)$ (broken) to $p_1(z)$ (target) using a U-Net parameterization of the velocity field, $\mathbf{u}_t = \frac{dz_t}{dt}$. **Right:** During inference, inputs with missing modalities are encoded and aligned through the learned flow, enabling consistent reconstruction or synthesis of complete modalities.

likelihoods. Latent Flow Matching (LFM) (Dao et al., 2023; Chadebec et al., 2025) extends FM by introducing a lower-dimensional latent space $\mathcal{Z}$. An encoder $E_\phi : \mathcal{X} \to \mathcal{Z}$ maps data into this latent space, where a transport map is learned, *i.e.* $\mathbf{z} := E_\phi(\mathbf{x})$.

Let $\pi_0, \pi_1$ be distributions over the latent space $\mathcal{Z} \subseteq \mathbb{R}^d$. Define a time-dependent velocity field $v : \mathcal{Z} \times [0, 1] \to \mathbb{R}^d$ and consider a stochastic process $(\mathbf{z}_t)_{t \in [0,1]}$ governed by the stochastic differential equation (SDE):

$$d\mathbf{z}_t = v(\mathbf{z}_t, t)dt + \sigma(\mathbf{z}_t, t)dB_t, \quad \mathbf{z}_0 \sim \pi_0, \tag{2}$$

where $B_t$ is standard Brownian motion and $\sigma$ is the diffusion coefficient. In practice, $v$ is approximated using the analytically defined target velocity field:

$$v^*(\mathbf{z}_t, t) := \frac{\mathbf{z}_1 - \mathbf{z}_t}{1 - t}, \quad (\mathbf{z}_0, \mathbf{z}_1) \sim \pi_0 \times \pi_1, \tag{3}$$

with intermediate states $\mathbf{z}_t$ sampled from the stochastic interpolant (deterministic when $\sigma \to 0$):

$$\mathbf{z}_t = (1 - t)\mathbf{z}_0 + t\mathbf{z}_1 + \sigma\sqrt{t(1-t)}\epsilon, \quad \epsilon \sim \mathcal{N}(0, I). \tag{4}$$

A neural velocity field $v_\theta(\mathbf{z}, t)$ is trained to match $v^*$ by minimizing the following error:

$$\mathcal{L}_{\text{LFM}}(\theta) = \mathbb{E}_{t, \mathbf{z}_0, \mathbf{z}_1, \epsilon}\left[\left|v_\theta(\mathbf{z}_t, t) - \frac{\mathbf{z}_1 - \mathbf{z}_t}{1 - t}\right|^2\right]. \tag{5}$$

Once trained, the learned velocity field $v_\theta$ is used to integrate the SDE in Eq. equation 2, transporting samples from $\mathbf{z}_0$ toward $\mathbf{z}_1$. The final outputs in the original data space are obtained by decoding the transported latent samples: $\mathbf{x}_1 := D_\psi(\mathbf{z}_1)$.

### 3.2 FLOWMI: FLOW-BASED MISSING MODALITY IMPUTATION

Existing multimodal models often handle missing modalities by simply substituting zeros or noise, which yields semantically meaningless inputs and degrades performance.

In contrast, we propose **FlowMI**, which treats missing-modality imputation as a latent-space reconstruction problem. In our framework, a multi-modal autoencoder projects both complete and incomplete inputs into a shared latent space. An input with missing modalities produces a broken latent code (due to the absent information), and we learn a continuous flow in latent space to transform this broken code into a corresponding full latent code as if all modalities were present. We train this latent transformation via a flow-matching objective that aligns broken latents with their ground-truth complete counterparts, enabling accurate reconstruction under arbitrary missing patterns. We detail the components of FlowMI below.

**Problem Setup.** Consider a multi-modal input $\mathbf{x} = \{x^{(1)}, x^{(2)}, \ldots, x^{(M)}\}$ consisting of $M$ modalities. Due to data missingness, only a subset of these modalities may be available at inference time. We represent the observed pattern with a binary mask $\mathbf{m} \in \{0,1\}^M$, where $\mathbf{m}^{(i)} = 1$ indicates modality $i$ is present and $\mathbf{m}^{(i)} = 0$ indicates it is missing. Given a mask $\mathbf{m}$, let $\mathbf{x}^{\mathbf{m}}$ denote the set of observed modalities and $\mathbf{x}^{\neg\mathbf{m}}$ the set of missing modalities for input $\mathbf{x}$. The goal of imputation is to predict the missing components $\mathbf{x}^{\neg\mathbf{m}}$ from the observed ones $\mathbf{x}^{\mathbf{m}}$. For notational convenience, let $\mathbb{1} = \{1, 1, \ldots, 1\}$ denote the mask of all ones (all modalities present), so $\mathbf{x}^{\mathbb{1}}$ is a fully observed input.

**Latent Representation.** We adopt a multi-modal variational autoencoder (VAE) framework in which each modality has a dedicated encoder and all modalities share a common latent space. Formally, let $\mathcal{M} = \{1, 2, \ldots, M\}$ be the set of modality indices. For each $m \in \mathcal{M}$, the encoder $E_\theta^{(m)}$ produces an approximate posterior distribution over the latent $\mathbf{z}$ given that modality's input:

$$q_\theta^{(m)}(\mathbf{z} \mid \mathbf{x}^{(m)}) = E_\theta^{(m)}(\mathbf{x}^{(m)}). \tag{6}$$

When a subset of modalities $\mathcal{O} \subseteq \mathcal{M}$ is observed, we combine their encoders' outputs using a emphProduct-of-Experts (PoE) to obtain a unified latent posterior:

$$q_\theta(\mathbf{z} \mid \{\mathbf{x}^{(m)}\}_{m\in\mathcal{O}}) \propto p(\mathbf{z}) \prod_{m\in\mathcal{O}} q_\theta^{(m)}(\mathbf{z} \mid \mathbf{x}^{(m)}), \tag{7}$$

where $p(\mathbf{z})$ is the prior distribution (i.e. $\mathcal{N}(0, I)$). Missing modalities are handled by simply omitting their encoder "expert" from the product. A shared decoder $D_\phi$ then maps latent codes back to the data space. Given a latent sample $\mathbf{z}$ drawn from the posterior $q_\theta$, the decoder produces a reconstruction for all modalities: $\hat{\mathbf{x}} = D_\phi(\mathbf{z})$.

During training, we randomly mask out modalities to create partial observations and optimize three loss terms. Given an observed subset $\mathcal{O}$ (with target modalities $\mathcal{T}$ to reconstruct, typically $\mathcal{T} = \mathcal{M}$), we sample $\mathbf{z}_\mathcal{O} \sim q_\theta(\mathbf{z} \mid x^{(m)} : m \in \mathcal{O})$ and decode it to $\hat{\mathbf{x}} = D_\phi(\mathbf{z}_\mathcal{O})$. We minimize the mean squared error between the reconstruction and the ground truth target $\mathbf{x}^{\mathbb{1}}$:

$$\mathcal{L}_{\text{rec}} = \mathbb{E}_{\mathbf{x}, \mathcal{O}} \left[ \left\| D_\phi(\mathbf{z}_\mathcal{O}) - \mathbf{x}^{\mathbb{1}} \right\|_2^2 \right]. \tag{8}$$

To encourage coherence between partial-input latents and full-input latents, we introduce a alignment penalty that brings the two posterior distributions, the broken latent $\mathbf{z}^{\mathbf{m}}$ (from incomplete inputs) and the full latent $\mathbf{z}^{\mathbb{1}}$ (from complete inputs), closer. Using stop-gradient on $\mathbf{z}^{\mathbb{1}}$, we minimize:

$$\mathcal{L}_{\text{pull}} = \mathbb{E}_{\mathbf{x}, \mathbf{m}} \left[ \|\mathbf{z}^{\mathbf{m}} - \text{sg}(\mathbf{z}^{\mathbb{1}})\|_2^2 \right], \tag{9}$$

where $\text{sg}(\cdot)$ denotes stop-gradient.

We also regularize the each modality-specific latent distribution against the prior $p(\mathbf{z})$ (as in a standard VAE):

$$\mathcal{L}_{\text{KL}} = \sum_{m=1}^{M} \mathbb{E}_{x^{(m)}} \left[ D_{\text{KL}}\big(q_\theta^{(m)}(\mathbf{z}^{(m)} \mid x^{(m)}) \parallel p(\mathbf{z})\big) \right]. \tag{10}$$

The full objective combines all three terms:

$$\mathcal{L} = \mathcal{L}_{\text{rec}} + \lambda\mathcal{L}_{\text{pull}} + \beta\mathcal{L}_{\text{KL}}, \tag{11}$$

where $\lambda$ and $\beta$ control the relative contributions.

**Latent Flow Matching.** The key component of FlowMI is a learned latent-space transformation that maps a broken latent code to its complete counterpart. We define a time-dependent vector field $v_\theta(\mathbf{z}, t)$ (implemented by a neural network) which generates a continuous trajectory from the broken latent to the full latent. Specifically, let $\mathbf{z}_0 = \mathbf{z}^{\mathbf{m}}$ be the latent obtained from an incomplete input (with mask $\mathbf{m}$), and let $\mathbf{z}_1 = \mathbf{z}^{\mathbb{1}}$ be the latent of the same input if all modalities were present. We define an ordinary differential equation (ODE) in latent space:

$$\frac{\mathrm{d}\mathbf{z}_t}{\mathrm{d}t} = v_\theta(\mathbf{z}_t, t), \quad \text{with} \quad \mathbf{z}_0 := \mathbf{z}^{\mathbf{m}}, \quad \text{and} \quad \mathbf{z}_1 := \mathbf{z}^{\mathbb{1}}, \tag{12}$$

To make learning the flow tractable, we prescribe a simple path and train $v\theta$ to follow it. In particular, we use the straight-line interpolation between the endpoints as the target trajectory: $\mathbf{z}_t = (1-t)\mathbf{z}_0 + t\mathbf{z}_1$. Along this path, the true velocity is constant and given by $\mathrm{d}\mathbf{z}_t/\mathrm{d}t = \mathbf{z}_1 - \mathbf{z}_0$. We train the vector field $v\theta$ to match this velocity at every point along the path via a flow-matching loss:

$$\mathcal{L}_{\text{LFM}} = \mathbb{E}_{(\mathbf{z}_0,\mathbf{z}_1),t\sim\mathcal{U}(0,1)} \left[ \left\| v_\theta\left(\mathbf{z}_t, t\right) - (\mathbf{z}_1 - \mathbf{z}_0) \right\|^2 \right] \tag{13}$$

In essence, the learned flow function $v_\theta$ provides an efficient latent-space imputation dynamics that can handle arbitrary missing modality patterns.

**Inference.** At test time, given an incomplete input $\mathbf{x}^{\mathbf{m}}$ with mask $\mathbf{m}$, we first obtain its broken latent code via the encoders: $\mathbf{z}^{\mathbf{m}} = E_\theta(\mathbf{x}^{\mathbf{m}})$. We then apply the learned latent flow to transform $\mathbf{z}^{\mathbf{m}}$ toward an estimate of the full latent. Starting from $\mathbf{z}_0 = \mathbf{z}^{\mathbf{m}}$, we numerically integrate the ODE $\mathrm{d}\mathbf{z}_t/\mathrm{d}t = v_\theta(\mathbf{z}_t, t)$ from $t = 0$ to $1$. For example, using a simple Euler integration with step size $\Delta t$, we update the latent as:

$$\widehat{\mathbf{z}}_{t+\Delta t} = \mathbf{z}_t + \Delta t \cdot v_\theta(\mathbf{z}_t, t), \quad \text{for } t = 0, \Delta t, 2\Delta t, \dots, 1 - \Delta t. \tag{14}$$

After integrating to $t = 1$, we obtain an approximate full latent $\hat{\mathbf{z}}^{\mathbb{1}} = \mathbf{z}_1$. Finally, we feed this latent into the decoder to reconstruct the complete input: $\hat{\mathbf{x}}^{\mathbb{1}} = D_\phi(\hat{\mathbf{z}}^{\mathbb{1}})$. The output $\hat{\mathbf{x}}^{\mathbb{1}}$ is the imputed multi-modal data, in which all originally missing modalities have been filled in by FlowMI method.

## 4 EXPERIMENTS

### 4.1 BASELINE SETUP

We evaluate and benchmark four categories of models for multimodal translation and missing-modality synthesis on PariedContrast: **Direct methods** (UNet (Ronneberger et al., 2015), ResViT (Transformer) (Dalmaz et al., 2022), MambaIR (Guo et al., 2024), I2IMamba (Atli et al., 2024), and RestoreRWKV (Yang et al., 2024)); **GAN-based methods** (CycleGAN (Zhu et al., 2017) and Pix2Pix (Isola et al., 2017)); **Diffusion-based methods** (PatchDiff (UNet) (Özdenizci & Legenstein, 2023) and DiTSR (Transformer) (Cheng et al., 2025)); **Flow-matching methods** (ConcatFM (Lipman et al., 2022), DirectFM (Lipman et al., 2022), PMRF (Ohayon et al., 2024), and our proposed FlowMI).

Training uses direct supervision on explicitly paired CE–NCE scans. All methods are trained under the same data split and preprocessing pipeline (Sec. 2.2), without external data. We follow each method's public implementation for loss functions and training protocols. Training is performed on 2D slices from registered volumes, resized to a fixed resolution. *Additional implementation details and hyperparameters are provided in the Appendix.* We report results using PSNR and SSIM as quantitative metrics.

### 4.2 EXPERIMENT RESULTS

**Quantitative comparison.**

Tab. 3 shows that flow-matching methods consistently outperform direct, GAN, and diffusion baselines. Across all tasks, our model achieves the highest PSNR and competitive SSIM, with a clear lead on CT→CTC, demonstrating its ability to generate both sharp and structurally faithful reconstructions. We observe that $n \to 1$ translation generally benefits from complementary temporal information, whereas $1 \to n$ remains the most challenging due to the difficulty of preserving structural consistency across multiple targets. Overall, diffusion models yield stable but moderate results, and state-space approaches improve SSIM by capturing temporal enhancement patterns. Flow-matching provides robust gains across both PSNR and SSIM.

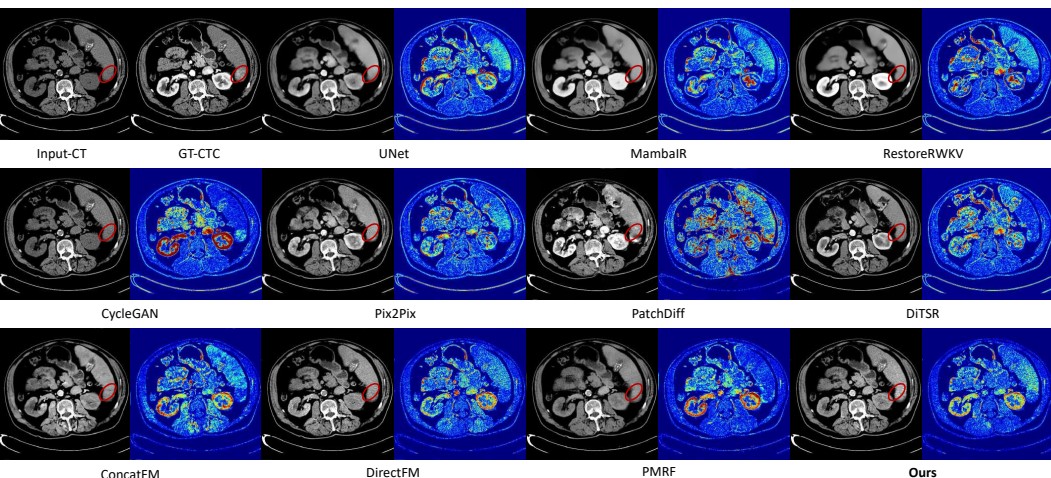

Figure 4: CT→CTC (liver). Red circles mark tumor regions. Input CT shows no clear lesion, ground-truth CTC shows bright enhancement. Most methods under-enhance the tumor, while PatchDiff and ours recover the correct signal. Alongside each result, the blue residual maps visualize differences from ground truth (darker indicates larger error).

Table 3: Quantitative comparison on the PairedContrast CT and MR paired pan-cancer contrast media dataset using methods grouped by generative mechanism and architecture. *Trans: Transformer.

| Mechanism | Architecture | Methods | CT → CTC | | DCE$_1$ → DCE$_2$ | | DCE$_1$ → DCE$_{2,3}$ | | DCE$_{1,3}$ → DCE$_2$ | |
|---|---|---|---|---|---|---|---|---|---|---|
| | | | PSNR | SSIM | PSNR | SSIM | PSNR | SSIM | PSNR | SSIM |
| Direct | UNet | UNet (Ronneberger et al., 2015) | 22.64 | 0.7726 | 24.36 | 0.7160 | 24.78 | 0.7720 | 27.54 | 0.7575 |
| | Trans | ResViT (Dalmaz et al., 2022) | 20.80 | 0.7419 | 25.12 | 0.7055 | 24.65 | 0.6347 | 25.24 | 0.5983 |
| | Mamba | MambaIR (Guo et al., 2024) | 23.88 | 0.7643 | 25.61 | 0.6756 | 26.12 | 0.7777 | 28.78 | **0.7889** |
| | Mamba | I2IMamba (Atli et al., 2024) | 20.97 | 0.7456 | 23.25 | 0.5588 | 23.57 | 0.5798 | 26.82 | 0.7006 |
| | RWKV | RestoreRWKV (Yang et al., 2024) | 23.58 | 0.7794 | 26.32 | **0.7748** | 26.29 | **0.7969** | 25.97 | 0.7100 |
| GAN | UNet | CycleGAN (Zhu et al., 2017) | 21.90 | 0.7579 | 24.18 | 0.6566 | 24.46 | 0.7137 | 25.74 | 0.7237 |
| | UNet | Pix2Pix (Isola et al., 2017) | 21.39 | 0.7264 | 23.24 | 0.6462 | 23.90 | 0.7223 | 26.42 | 0.7015 |
| Diffusion | UNet | PatchDiff (Özdenizci & Legenstein, 2023) | 21.44 | 0.7295 | 24.25 | 0.7041 | 25.32 | 0.7579 | 27.53 | 0.7563 |
| | Trans | DiTSR (Cheng et al., 2025) | 22.68 | 0.7612 | 25.59 | 0.7556 | 25.69 | 0.7645 | 27.50 | 0.7573 |
| Flow-matching | UNet | ConcatFM (Lipman et al., 2022) | 23.10 | 0.7712 | 26.31 | 0.7292 | 26.25 | 0.7177 | 29.06 | 0.7612 |
| | UNet | DirectFM (Lipman et al., 2022) | 22.84 | 0.7665 | 25.74 | 0.6848 | 25.98 | 0.7110 | 27.90 | 0.7415 |
| | UNet | PMRF (Ohayon et al., 2024) | 21.91 | 0.7656 | 25.06 | 0.6751 | 26.11 | 0.7012 | 27.53 | 0.7494 |
| | UNet | FlowMI (Ours) | **24.47** | **0.7846** | **26.52** | 0.7415 | **26.63** | 0.7369 | **29.17** | 0.7622 |

**Qualitative comparison.** As shown in Fig. 4, the qualitative results echo the quantitative findings. While most methods generate visually plausible CT images, they often leave tumor regions under-enhanced, making lesions difficult to distinguish. In contrast, PatchDiff and our flow-matching model successfully reproduce the bright enhancement seen in the ground-truth CTC, and the accompanying blue residual maps confirm lower reconstruction errors. These results highlight that beyond visual realism, clinically accurate contrast synthesis is essential for downstream diagnosis.

## 5 CONCLUSIONS

In this work, we introduced PariedContrast, a comprehensive pan-cancer benchmark for multi-modal image translation and missing-modality synthesis in clinically realistic settings. It provides high-quality, per-patient paired CT and MR scans across 11 organs with both contrast-enhanced and non-contrast modalities, curated through a standardized and reproducible pipeline. We defined three benchmark tasks—1-to-1, $n$-to-1, and $n$-to-$n$ translation—and reported reference results using representative generative models. While limitations remain, such as biases in public data sources and variability in clinical imaging, PariedContrast offers a valuable foundation for developing robust multimodal translation methods and advancing clinically reliable decision support. Future work could extend the dataset with additional modalities, larger cohorts, and more advanced synthesis techniques to further enhance generalization and applicability.

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

## A  ADDITIONAL DATASET DETAILS

Here we provide extended statistics for PariedContrast (see Tab. 1) as well as the original data sources of each collection.

All scans were obtained by downloading raw DICOM or NIfTI files from publicly available repositories. The included collections are: Adrenal-ACC-Ki67-Seg (Moawad et al., 2023) TCGA-OV (Holback et al., 2016), TCGA-UCEC (Erickson et al., 2016a), CPTAC-UCEC (Consortium et al., 2019), TCGA-STAD (Lucchesi & Aredes, 2016), CPTAC-PDA (, CPTAC), HCC-TACE-Seg (Moawad et al., 2021), TCGA-LIHC (Erickson et al., 2016b), CMB-CRC (Biobank, 2022b), TCGA-COAD (Kirk et al., 2016b), TCGA-BLCA (Kirk et al., 2016c), TCGA-KIRC (Akin et al., 2016), C4KC-KiTS (Heller et al., 2019), CPTAC-CCRCC (, CPTAC), TCGA-KIRP (Linehan et al., 2016b), TCGA-KICH (Linehan et al., 2016a), CMB-LCA (Biobank, 2022a), CPTAC-LSCC (Consortium et al., 2018a), CPTAC-LUAD (Consortium et al., 2018b), Lung-PET-CT-Dx (Li et al., 2020), TCGA-LUSC (Kirk et al., 2016a), Anti-PD-1_Lung (Madhavi et al., 2019), TCGA-BRCA (Lingle et al., 2016)(Wu et al., 2021), UCSF (Li et al., 2008)(Jafri et al., 2014)(Wu et al., 2021), I-SPY 1 (Newitt et al., 2016)(Wu et al., 2021).

## B  LLM USAGE

We employed large language models (LLMs) strictly as writing assistants. Their role was limited to grammar correction, wording refinement, and improving readability. All technical content, methodology, experiments, and analyses were conceived, implemented, and validated entirely by the authors. No results, data interpretations, or methodological decisions were generated by LLMs.

