# OpenReview forum: "PairedContrast: A Multimodal Benchmark for Medical Image Translation"
_ICLR.cc/2026/Conference — ICLR 2026 Conference Withdrawn Submission_

### Official Review · Reviewer_Zoaj · 2025-10-28

**Soundness:** 3
**Presentation:** 2
**Contribution:** 2
**Rating:** 4
**Confidence:** 4

**Summary:**

This paper introduces PairedContrast, a proposed pan-cancer benchmark for contrast-enhanced medical image translation across CT and MRI modalities. The dataset aggregates 2,642 paired scans from public repositories such as TCIA, TCGA, and CPTAC, covering 11 organs and 19 cancer types. The authors further propose FlowMI, a flow-matching–based imputation model designed to reconstruct missing modalities by transporting latent representations from incomplete to complete inputs. The benchmark evaluates GAN-, diffusion-, and flow-based baselines, reporting that FlowMI achieves the highest PSNR/SSIM.

**Strengths:**

1. Reasonably clear motivation and setup. The paper addresses a relevant problem in medical image translation, aiming to synthesize contrast-enhanced scans from non-contrast inputs, which is clinically meaningful.
2. Dataset construction effort. The authors collected and organized a multi-organ dataset with explicit pairing between contrast and non-contrast scans. While not groundbreaking, it provides a reasonably broad coverage compared to prior single-organ datasets.
3. The proposed FlowMI adapts latent flow-matching to address missing modalities via a straightforward latent-space interpolation scheme. Its implementation appears methodologically sound, and the evaluation framework incorporates comparisons against standard generative baselines, demonstrating favorable performance in the tested scenarios.

**Weaknesses:**

1. Lack of downstream or clinical validation. The paper presents a “pan-cancer” benchmark and claims clinical relevance, but does not evaluate downstream clinical tasks such as tumor segmentation, detection, or diagnosis. All results rely on PSNR and SSIM, which measure low-level reconstruction fidelity but not diagnostic or functional consistency. Without lesion- or outcome-level validation, the claimed clinical significance of contrast synthesis remains unsubstantiated.
2. Terminological and presentation inconsistencies. The manuscript alternates between “PariedContrast” and “PairedContrast”;
3. Missing baselines and insufficient insight. Several relevant multimodal imputation or translation baselines (e.g., MM-GAN[1], Seq2Seq[2]) are not included.
4. Incomplete methodological documentation. The paper states that it applies rigid and affine alignment followed by deformable registration for motion-prone organs such as the liver and lung, but it does not specify which algorithm or tool was used (e.g., ANTs, Elastix, or VoxelMorph), nor provide parameter settings. The accuracy of registration is crucial for image translation tasks, since misalignment between contrast-enhanced and non-contrast scans can introduce structural inconsistencies that directly affect model supervision and evaluation reliability.
5. No robustness, variance, or detail analyses. The experimental results report only mean PSNR and SSIM values across the test-mini split, without presenting standard deviations or per-organ variance. Given that PairedContrast includes 11 organs and multiple imaging modalities with inherently diverse characteristics, the absence of statistical reporting makes it difficult to evaluate the model’s stability and generalizability. For such a heterogeneous benchmark, per-organ variance analysis is essential to verify the robustness and reliability of the reported improvements.

## Reference
[1] Sharma A, Hamarneh G. Missing MRI pulse sequence synthesis using multi-modal generative adversarial network[J]. IEEE transactions on medical imaging, 2019, 39(4): 1170-1183.

[2] Han L, Tan T, Zhang T, et al. Synthesis-based imaging-differentiation representation learning for multi-sequence 3D/4D MRI[J]. Medical Image Analysis, 2024, 92: 103044.

**Questions:**

1. Have you considered downstream evaluations such as segmentation or lesion detection to demonstrate the clinical validity of your synthesis?
2. Have you obtained formal permissions or legal clearance to redistribute derived datasets aggregated from TCIA/TCGA/CPTAC sources?
3. Which algorithm or software did you use for the deformable registration step (e.g., ANTs, Elastix, MONAI Core)? Please cite it.
4. Where exactly in the Appendix can readers find the claimed table (from section 2.3, "A detailed breakdown of case numbers per organ, modality, and cancer type is provided in Appendix.")? If missing, please include it.

---

### Official Review · Reviewer_aBmj · 2025-10-29

**Soundness:** 1
**Presentation:** 1
**Contribution:** 1
**Rating:** 2
**Confidence:** 4

**Summary:**

The paper presents three main contributions:
1. A dataset comprising paired non-contrast and contrast-enhanced CT and MR scans covering diverse anatomical sites in oncology cases.
2. A latent flow matching framework designed to generate contrast-enhanced images from non-contrast scans.
3. A benchmark comparing the proposed approach with existing state-of-the-art methods on the newly introduced dataset.

**Strengths:**

- The effort to assemble a dataset enabling the development and evaluation of methods for synthesising contrast-enhanced images from non-contrast scans is valuable.
- The proposed latent flow matching approach demonstrates reasonable quantitative performance.
- The approach is evaluated against a broad set of diverse baselines on a relatively large and heterogeneous dataset.

**Weaknesses:**

### General
- The overall positioning of the paper is unclear. If the main focus is the dataset, the description lacks sufficient detail; if it is the latent flow matching method, the novelty is not convincingly demonstrated; and if it is the benchmark, the experimental analysis is not strong enough.
- The literature review and contextualisation are very limited. Not a single prior work on synthesising contrast-enhanced images from non-contrast scans is cited, despite several relevant studies, for instance:
    1. Chen C, Raymond C, Speier W, et al. Synthesizing MR Image Contrast Enhancement Using 3D High-resolution ConvNets. IEEE Transactions on Biomedical Engineering. Published online 2022:1-12. doi:10.1109/TBME.2022.3192309
    2. Xu C, Zhang D, Chong J, Chen B, Li S. Synthesis of Gadolinium-enhanced Liver Tumors on Nonenhanced Liver MR Images Using Pixel-level Graph Reinforcement Learning. Medical Image Analysis. 2021;69:101976. doi:10.1016/j.media.2021.101976
    3. Bône A, Ammari S, Lamarque JP, et al. Contrast-enhanced brain MRI synthesis with deep learning: key input modalities and asymptotic performance. In: International Symposium on Biomedical Imaging. 2021. Accessed February 23, 2021. https://hal.archives-ouvertes.fr/hal-03128023
    4. Jayachandran Preetha C, Meredig H, Brugnara G, et al. Deep-learning-based synthesis of post-contrast T1-weighted MRI for tumour response assessment in neuro-oncology: a multicentre, retrospective cohort study. The Lancet Digital Health. 2021;3(12):e784-e794. doi:10.1016/S2589-7500(21)00205-3
    5. Gong E, Pauly JM, Wintermark M, Zaharchuk G. Deep learning enables reduced gadolinium dose for contrast-enhanced brain MRI. Journal of Magnetic Resonance Imaging. 2018;48(2):330-340. doi:https://doi.org/10.1002/jmri.25970
    6. Piening M, Altekrüger F, Steidl G, Hattingen E, Steidl E. Conditional Generative Models for Contrast-Enhanced Synthesis of T1w and T1 Maps in Brain MRI. arXiv. 2024. doi:10.48550/arXiv.2410.08894
- The paper is poorly written overall, with multiple spelling and grammatical errors, including the dataset name apparently misspelled throughout the text, even in the title.

### Proposed dataset
- The paper does not clarify whether the source datasets permit redistribution, casting doubt on whether the final assembled dataset can be legally shared. There is no mention of where the assembled dataset can be found.
- There is no information on demographic diversity, scanner types, or acquisition protocols, leaving the representativeness and generalisation potential of the dataset uncertain.
- Preprocessing steps are insufficiently described, particularly the registration procedure.

### Proposed method
- Due to the limited review of related work, the methodological novelty and contribution of the latent flow matching approach remain unclear.

### Experiments
- Results appear to be based only on the “test-mini” split (approximately 5% of the dataset, or around 130 images), which is a limited sample size for robust evaluation.
- No measures of variability (e.g., standard deviation or confidence intervals) are provided, preventing meaningful comparison between methods.

**Questions:**

- What is the main focus of the paper — the dataset, the proposed latent flow matching method, or the benchmarking framework?
- How does your approach differ from prior work on contrast-enhanced image synthesis?
- Can you clarify the novelty of the latent flow matching method compared to standard latent diffusion or flow-based models?
- Are you legally allowed to redistribute the assembled dataset?
- Can you provide details on dataset diversity (demographics, scanner types, acquisition protocols)?
- Why are results reported only on the small test-mini split (~5% of data)?
- Could you include measures of variability (e.g., standard deviations or confidence intervals) to better support the reported quantitative results?

**Details Of Ethics Concerns:**

The authors claim or imply a public release but do not provide evidence that redistribution complies with the terms of use of the original datasets.

---

### Official Review · Reviewer_KT9Y · 2025-11-01

**Soundness:** 2
**Presentation:** 2
**Contribution:** 2
**Rating:** 4
**Confidence:** 3

**Summary:**

The authors propose a new dataset, PairedContrast, which is a fully paired dataset spanning 11 human organs across various pan-cancer medical imaging modalities. Apart from the dataset, they propose a missing modality data augmentation technique-FlowMI, inspired by flow matching. They evaluate the efficacy of FlowMI on the task of correct annotation generation against a diverse set of similar annotation techniques. The results suggest that FlowMI is a plausible annotation technique for missing modalities.

**Strengths:**

- Use of Flow matching as an annotation tool is an interesting idea.

- Release of a novel, large-scale, fully paired dataset, a pan-cancer medical imaging dataset spanning 11 human organs.

- The paper is well written with proper definitions, and clear figures and tables, improving the overall understanding.

**Weaknesses:**

- No details regarding the reproducibility of the training method. Eg. Size of the NN for flow matching velocity prediction, learning rate of the VAE training, number of data points used for training, which specific encoders for each of the modalities were used for the VAE and why, etc. The submission would have benefited from some supporting source code or discussion about the implementation details in the appendix. There are no implementation details mentioned in the appendix regarding experiments, as promised in section 4.1 (line 419)

- The evaluations only cover Distortion metrics like PSNR and SSIM, and provide examples of sample images for highlighting the quality of the generated images. It would be better to see some quantitative results for qthe uality of the generated images using some perpetual quality metrics like FID, KID, etc. scores.

- The dataset preprocessing task involves human-in-the-loop with trained annotators(section 2.2, line 189). There is no discussion about the consent and ethical consideration or the expert level of the annotators. It would be useful to indicate the level of the experts for building the confidence of someone using the dataset in future.


- Presentation: (1) Spelling mistakes: 2.1 Task Definition(95), (2) Try to introduce the full forms when you use the short forms first(103-104), and (3) Increase the separation between the tables and the text.

**Questions:**

- Was FlowMI used in the curation of the current dataset? If yes, did the authors indicate how many data points were artificially annotated in the dataset, and a dedicated mark on annotated data points?

- Could the authors kindly clarify if they trained the U-net for Flow Matching in asynchronous with the VAEs? Or did they train them separately?

**Details Of Ethics Concerns:**

The authors release the dataset as a combination of multiple other datasets. These data points are sensitive and related to human subjects, and therefore, the authors should include an ethics statement in the paper describing any licenses/permissions associated with any datasets which were used.

---

### Note · Authors · 2025-11-14

**Comment:**

Thank you to the reviewers for their time and valuable feedback. We appreciate the comments and have decided to dedicate more effort to improving the work. Therefore, we are withdrawing the paper.

**Withdrawal Confirmation:**

I have read and agree with the venue's withdrawal policy on behalf of myself and my co-authors.